# Novel Dose of Natrium Chloride and Soil Concentration in Reducing Medical Waste Bacteria before Incineration

**Marsum Marsum [1,*] and Ismi Rajiani [2,*]**

1  Department of Public Health, Poltekkes Kemenkes, Semarang 50286, Indonesia
2  Department of Social Studies, Faculty of Teacher Training and Education, Lambung Mangkurat University, Banjarmasin 70123, Indonesia
*  Correspondence: marsumrahma1963@gmail.com (M.M.); rajiani@ulm.ac.id (I.R.)

**Abstract:** If it is not adequately managed, the waste from healthcare facilities containing infectious material poses a risk to the general public and the natural environment. As a result, hospitals must ensure that their waste management policies do not add to the dangers posed to both human health and the environment. In this study, we aimed to determine the effect that varying doses of disinfectant in conjunction with andosol soil had on the total number of bacteria present in the medical waste generated by three hospitals in Semarang City, Indonesia. According to the findings of the study, the most efficient method for decreasing the overall number of microbial colonies by 93% was a combination involving soil (at a percentage of 30) and chlorine (at a concentration of 0.75 ppm). As a consequence of this, and due to the limited technology available, this straightforward method can become an alternative for the healthcare industry in managing medical waste before dumping or incinerating it. Hospitals have been advised to discontinue the practice of directly burning, disinfecting, or transporting waste to disposal locations before it receives treatment. This can help reduce the risk of pandemics, as the correct disposal of medical waste can control infection sources.

**Keywords:** NaOCl; disinfectant; medical waste; bacteria; soil

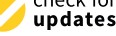



## 1. Introduction

A crisis has arisen on a global scale as a direct result of an unanticipated growth in the volume of waste, which has had a detrimental effect on the worldwide waste management industry and as a result has brought about a predicament that requires immediate attention. The global lockdown that was imposed as a result of the spread of COVID-19 had positive and negative impacts not only on the natural environment but also on other systems, such as those involved in garbage management. However, waste management is in its infancy in many developing countries; as a result, waste management during the pandemic period was severely impacted in many regions of the world.

Further, unpredictability has been acknowledged in both social and economic systems worldwide due to the COVID-19 outbreak. All facets of society have transformed, including the institutionalization of social distance, the reorganization of the public health system, and the manufacturing of hygiene products [1]. Although it has been reported that COVID-19 has had a positive impact on the environment, such as clear skies, cleaner rivers, clean beaches, and improvements in air quality [2], there is also apprehension regarding the harmful effects related to an increase in the amount of solid waste and a decrease in the number of recycling efforts [3]. Currently, one of the most challenging environmental issues is the administration of medical waste (MW), which consists of a variety of potentially infectious resources that can pose potential risks to the health of the general population [4]. Inadequate waste management in underdeveloped nations increases the likelihood of people being exposed to the coronavirus [5]. Because of this, the safe handling of medical waste and the final disposal of this waste is an essential component of a reactive response [6].

Up to this point, the most common methods for disposing of MW waste in underdeveloped nations have been dumping it into landfills, microwaving, using chemical disinfectants, incinerating, shredding, compacting, and using steam sterilization [6]. On the other hand, using these methods results in a number of adverse side effects. For instance, certain poisonous substances may seep into the clay or underground water in landfills, and conventional chemicals and enormous masses of waste cannot be disposed of using a microwave [4]. Throughout the past few decades [7], the advantages and drawbacks of these MW disposal methods have been the subject of debate. As a consequence of this, a different strategy is required in order to identify the method of MW disposal that is both effective and efficient.

The occurrence of COVID-19 cases in Indonesia induced a significant shift in the way in which garbage is generated and handled, which has had significant repercussions for the environment. After approximately two years, considerable increases in the amount of medical waste (such as face masks and personal protective equipment (PPE), gloves, goggles, and syringes) have been observed not only in hospitals but also in bodies of water. Both hospitals and trash banks have been overwhelmed by this enormous amount of rubbish. Using conventional methods it has become impossible to dispose of medical waste in an efficient manner [8]. Because of physical and social constraints, the general public and academia cannot participate in community service related to trash management [9]. As they collect, sort, and categorize recyclable garbage, members of informal worker groups, commonly known as waste collectors, are at an increased risk of developing adverse health disorders due to poor MW [10]. Presently, infected face masks, gloves, and tools for recognizing, spotting, and handling SARS-CoV-2 are becoming infectious garbage, which is an irreversible process. This will result in environmental and public health issues if the waste is improperly placed, transported, and stored. The threat is higher in developing economic countries that require more satisfactory methods of medical waste management. Economically developing countries in Asia, such as Cambodia, the Philippines, Thailand, India, Malaysia, Indonesia, Bangladesh, Vietnam, and Palestine, have commonly been observed to dump waste in poorly managed and open landfills [11]. Therefore, one of the problems that will inevitably occur is the presence infectious waste, which may be a source of severe diseases and environmental harm.

Additionally, a significant quantity of MW, including discarded gloves, face masks, disposable aprons and PPE, syringes, and other items, was produced at hospitals and treatment centers due to the COVID-19 outbreak. The unanticipated and urgent problem brought about by the current conditions has left waste management authorities needing clarification. In the context of developing nations, where many people do not have access to adequate education, ignorance among specific sectors of society leads to the haphazard dumping of the wastes mentioned above. The careless disposal of garbage in landfills, streets, or water bodies, among other places, may lead to severe repercussions for human health because the waste in question may be infectious and may still have the potential to carry residues of viral contaminants. Therefore, effective waste management has emerged as a significant obstacle for local waste management authorities, particularly in countries that are still in the development stage. This fear has been made worse by reports stating that the novel coronavirus could remain on the surfaces of objects that are used daily.

As a consequence of this, the biomedical wastes that are produced by hospitals and COVID-19 treatment centers, such as infectious and discarded masks, needles, syringes, bandages, gloves, used tissue, leftover medicines, etc., should be disposed of appropriately to minimize the further spread of COVID-19 and the contamination of the environment. Since dumping into the soil and the use of sodium hypochloride (NaOCl) are widely practiced in developing economic countries [12], the purpose of this work was to estimate the appropriate doses of NaOCl and soil concentrations required to reduce bacteria as part of self-management efforts by health service facilities to reduce the risk of medical waste contamination before incineration. It has been demonstrated that soil, mainly of the andosol type, can lessen the number of *B. cereus* bacteria present in syringe waste [13].

However, research has yet to be conducted into whether or not this soil is effective when combined with different concentrations of the disinfectant NaOCl. Andosols are soils from volcanically active regions. They possess soil characteristics that distinguish them from other soils. Andosol is formed from the Japanese words 'an', meaning dark, and 'do', indicating soil. When environmental conditions promote their development, andosols can also be found outside of active volcanic zones. Andosols cover a small portion (1–2%) of the Earth's land surface, yet are densely populated. We chose to study syringes because they are smaller but harder than other forms of medical waste such as discarded gloves, face masks, disposable aprons, and PPE. The proper doses for other forms of MW can be estimated if the appropriate amount can be identified for small but hard objects.

## 2. Materials and Methods

In this study, we used a pre-test design, followed by a post-test design. The pre-test was a test for the group of Gram-positive spore bacteria acquired from the isolation of the syringe bacteria, and the post-test was a test for *B. cereus* bacteria. To study the isolated bacteria, a suspension was created with a microbial fuel cell (MFC) density of 0.5. The isolate was subjected to a dilution of 0.85% NaOCl with as much as 1000 mL for 24 h. The working method involved submerging sterile syringes in a suspension for eight hours to produce a biofilm on the surface of the needle with pH 7 and a brain heart infusion (BHI) nutrient substrate. This step was performed so that the process could be carried out. This methodology adhered to the protocols followed by earlier researchers [14,15].

Andosol soil was extracted from the highlands and utilized in the treatment. The soil was sieved with a 3 mesh sieve to obtain a particle size of 6730 microns, and it was then dried at a temperature of 1100 degrees Celsius for 12 h. At a pH of 6, sterile distilled water creates soil with a concentration of thirty per cent. This followed a recent study that found that the chemical composition of the soil, namely, the levels of Fe (levels of 11.772 ppm), Zn (levels of 0.105 ppm), and Cu, were able to lower the number of bacteria in the soil (to levels of 0.018 ppm). In addition, microorganisms could be removed from the soil using an autoclave set at 1210 degrees Fahrenheit for thirty minutes [16].

Syringes that had been contaminated with *Bacillus cereus* bacteria for 8 h were treated with a different combination sequence, namely, NaOCl (30 min), 30% andosol soil (2 min). This treatment was carried out three times (30 min). The exposure time was based on the duration used in previous studies when conducting experiments with andosol soil [17,18]. In the first stage of the procedure, the syringe was submerged in a solution with a concentration that varied between 0.75 ppm, 0.375 ppm, and 0.188 ppm for thirty minutes. After that, a concentration of 0.094 ppm and 0.047 ppm of each were added to the soil suspension, and vice versa. The treatment with the NaOCl solution required a contact period of 30 min, but the andosol soil only required 2 min. The soil solution was brought to a uniform consistency by shaking it with an orbital shaker at 100 revolutions per minute. The sample from the syringe was placed in NaOCl, and the culture was carried out at 37 degrees Celsius for a full day. The final step was to compute the colony amount using the total plate count (TPC) method, which involved multiplying the number of bacteria present in a Petri dish by one and then dividing that result by the dilution factor [19]. To determine the best composition, one-way ANOVA was employed by examining the highest partial eta squared and values below 0.05 were considered significant.

Sample replication was calculated with the Federer formula: $(t - 1)(r - 1) \geq 15$, which is commonly used in medical research to determine samples [20]. Since there were 13 treatments, the calculation was as follows:

$$(13 - 1)(r - 1) \geq 15$$
$$12(r - 1) \geq 15$$
$$12r \geq 15 + 12$$
$$r = 27/12$$
$$r = 2.3 \sim 3$$

The number of replicates (the number of samples) was at least 3, but in this study we used a total sample number of 10. Thus, the total sample number for this study was 13 treatments multiplied by 10 = 130 samples (total samples).

## 3. Results and Discussion

In Table 1 we present the andosol soil mineral material composition, which consisted of lead (Pb), zinc (Zn), copper (Cu), and iron (Fe) (Pb).

**Table 1.** Soil chemical examination.

| Mineral Content | Level |
|---|---|
| Fe | 11.772 ppm |
| Zn | 0.105 ppm |
| Cu | 0.018 ppm |
| Pb | 0.006 ppm |

Previous studies [18] have investigated the possibility that soil can be used to reduce the number of bacteria in a given environment. Soil contains metallic minerals such as Ag, Al, Cu, Fe, Mg, Mn, Ni, and Zn, which have the potential to be antibacterial agents against *Salmonella typhi* and *Staphylococcus aureus*. Cu was detected in the soil in this investigation. Cu is toxic to bacteria upon direct contact, with the Cu process causing damage to the outer membrane of bacteria as a result of the presence of Cu ions. Copper's inactivation of the iron sulphur group damages the bacterial cell membrane's essential catabolic and biosynthetic pathways, as stated in [21]. Therefore, as an alternative to acquiring adequate and ecologically friendly technology to combat bacterial resistance, it can be employed as a solution in developing countries such as Indonesia. The use of soil can decrease the number of bacteria in an area. This is because zinc in soil has been shown to lower the number of *Escherichia coli* bacteria by 98% after 2 h and *Pseudomonas aeruginosa* colonies by 100% after a contact time of 4 h [7]. Due to the charge of ferric ions of iron ($Fe^{3+}$), the iron content in soil has antibacterial capabilities, which can be employed to lower the amount of *Escherichia coli* in secondary waste [20]. Bacteria can be eliminated through the bacteriostatic process that occurs in soil (bactericidal). When clay minerals kill bacteria, this process is referred to as a bactericidal process, whereas stopping bacterial growth and replication is referred to as a bacteriostatic process [22].

Initially, ferrous ion ($Fe^{2+}$) produced by ferrous iron metal is combined with oxygen in water to form iron oxide ($Fe(OH)_3$) and iron oxide ($Fe(OH)_3$) ($Fe_2O_3$). Secondly, the iron oxide produced can bind to bacteria through adsorption. This method can minimize the number of microorganisms in the water. Additionally, ferrous iron metal can generate hydroxyl free radicals ($OH^-$). Through oxidation, these radicals can harm the bacterial cell membrane. This damage can lead to the loss of ions and chemicals required by bacteria, preventing their reproduction or even causing them to perish. Fourth, persistent cell injury can result in bacterial inactivation or death. However, it should be emphasized that the stages of bacterial cell destruction can vary depending on iron ion concentration, pH, and other environmental factors [23,24].

Physical and chemical features of soil can inhibit bacterial populations. The minerals Al, Mg, Fe, and Si possess chemical properties that aid in lowering the amount of bacteria. This mineral improves the cation exchange capacity, also known as CEC; this ion exchange is fatal to bacteria. Clay contains minerals in the form of $Fe^{2+}$, which can infiltrate the protein structure of bacterial cells and generate hydroxide radials that can kill bacteria. The physical features of a porous/cavity-filled soil will lead germs to be absorbed/trapped until they die. Due to the sharp form of soil particles, the friction between bacteria and soil particles damages the bacterial cell wall [25].

Sodium hypochlorite (NaOCl) have been the most commonly used NaOCl-based disinfectants for cleaning surfaces and medical equipment during the COVID-19 pandemic [26].

The findings of the metal analysis of the soil sample revealed that it included material in the form of iron, zinc, and copper, all of which could kill bacteria. Microorganisms such as *E. coli* and *Staphylococcus aureus* could be eliminated from a solution with a contact time of three hours if the solution contained copper at a concentration of forty parts per million (ppm), supporting previous research conducted in 2015 showing that at a concentration of 50 mg/mL and a contact time of 24 h, soil material including the metal components Ag, Cu, Zn, and Co was able to kill *E. coli* [27]. This was achieved with a contact time of 24 h. Cu metal can cause damage to the surface of microbial cells and has the potential to kill SAR-CoV-2 [22]. A surface that is covered with Cu metal can also destroy SAR-CoV-2. Based on previous results, we deduced that the metal content of the soil was not capable of killing bacteria to the greatest extent possible in previous studies. As a result, it is essential to achieve the best possible combination of materials with NaOCl. The function of the soil was exploited in the application studied here to enhance the ability of bacteria to absorb nutrients from the soil.

The presence of NaOCl compounds affects the mortality of bacteria because NaOCl possesses bactericidal capabilities. These properties manifest themselves through the formation of hypochloric acid and the release of chlorine compounds, both of which alter the morphology of *B. cereus* bacteria. Figure 1 displays the findings of the research that was conducted using NaOCl concentrations of 0.75 parts per million (ppm), 0.375 ppm, 0.1888 ppm, 0.094 ppm, and 0.047 ppm in the treatment of numerous different combinations. The effect of exposure to NaOCl at several concentrations with a duration of 30 min on the morphology of *Bacillus cereus* bacteria using fluorescence and Gram staining demonstrated that *Bacillus cereus* in NaOCl solution at a concentration of 0.047 ppm began to respond by forming more spores to react to stimuli from an unfavorable environment. At a concentration of 0.188, the response was found to be the strongest, with significantly increased spore production. When performing Gram staining using the conventional protocols, it is possible to see on samples that have not been exposed to light-blue NaOCl that the color absorption increases at concentrations ranging from 0.188 ppm to 0.75 ppm. The maximum concentration reached was 0.75 ppm. Because *B. cereus* is a spore-forming, Gram-positive, aerobic or facultative anaerobic bacterium, the existence of peritrichous ciliature (having a consistent distribution of flagella across the surface of the body) makes it possible for the bacteria to move [22]. Bacilli belonging to the *B. cereus* species have a morphology that can be described as either thin and straight or slightly curled [28].

The spores of *B. cereus* bacteria are not harmed by a chlorine concentration of 100 ppm [29], but a chlorine concentration of 100 ppm can cause damage to the vegetative cells of *B. cereus* bacteria. The ability of *B. cereus* bacteria to increase at a concentration of 250 ppm of NaOCL could be improved through the exposure of the bacteria to heat stress and chemicals [30]. As shown in Figure 1, the findings of this research indicated that a concentration of 0.75 ppm NaOCl can have the impact of increasing the total number of spores.

The application of a treatment consisting of NaOCl and soil produced a more beneficial effect. When it came to increasing the bacteria's ability to adhere to surfaces, NaOCl was more effective in slightly alkaline conditions, whereas the soil was more effective in conditions that were acidic [31]. Table 2 shows that the results regarding decreasing the number of bacteria from each treatment showed slight variations or were typically distributed ($p > 0.05$), except for the treatment with chlorine 0.09373 + 30% soil, which showed a significant deviation or was not normally distributed ($p = 0.003$). Table 2 also shows that the treatment with chlorine 0.09373 + 30% soil showed a considerable variation or was not normally distributed ($p = 0.003$).

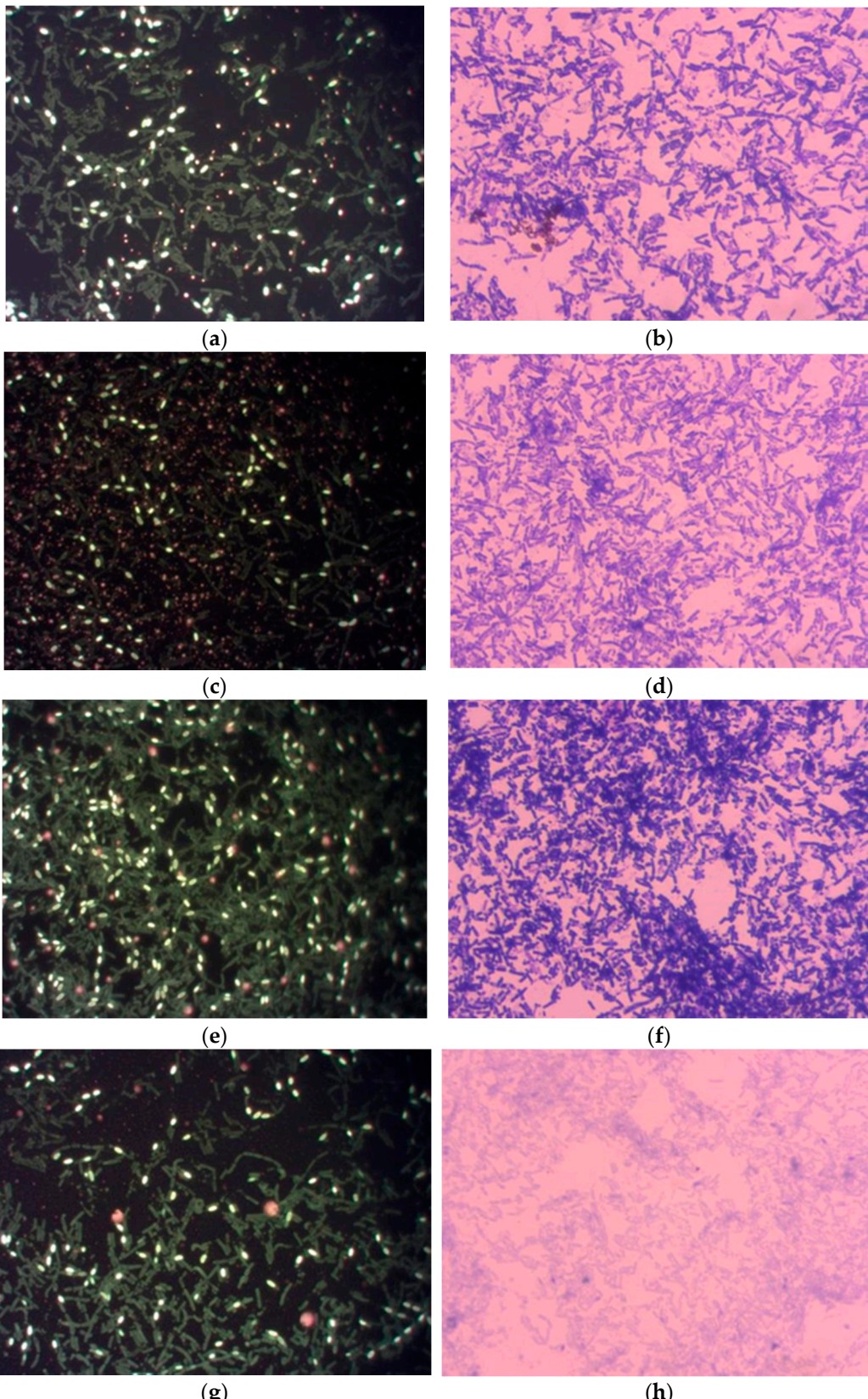

**Figure 1.** *Cont.*

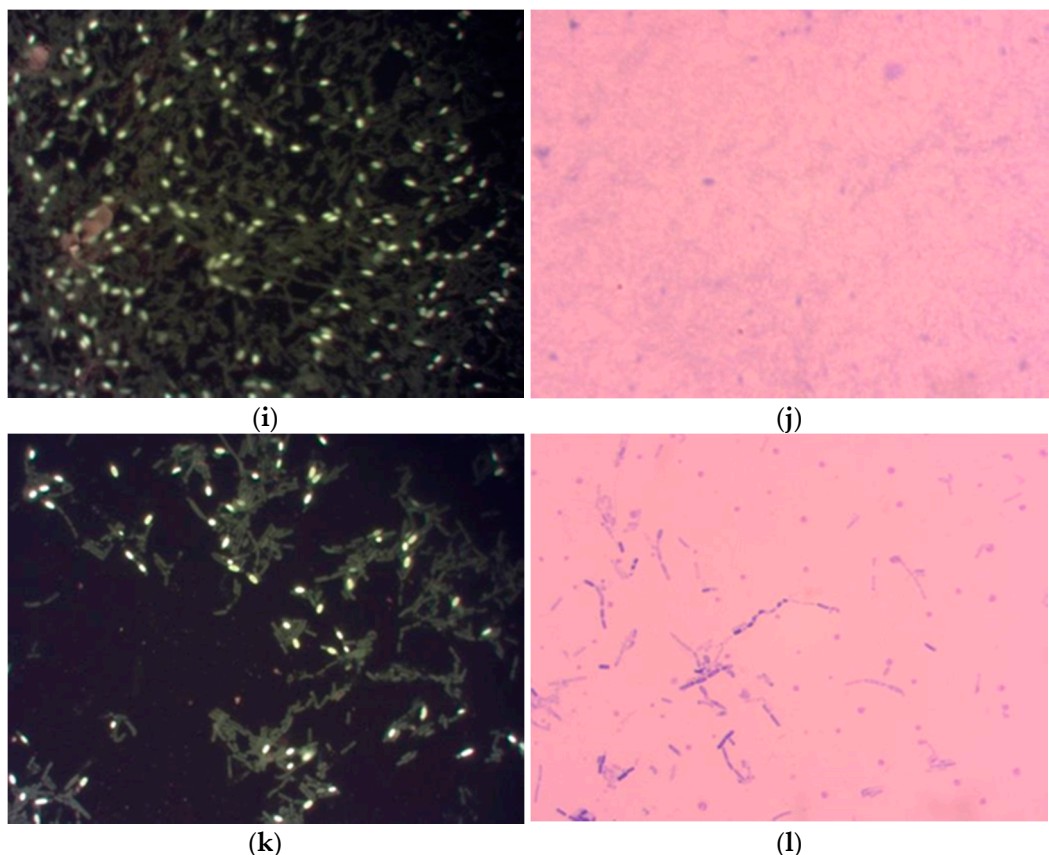

**Figure 1.** Bacterial morphology of *Bacillus cereus*, assessed with fluorescence and Gram staining. (**a,c,e,g,i**) fluorescent staining of the bacterial morphology at 0.75, 0.375, 0.188, 0.094, and 0.047 ppm NaOCL concentrations, respectively. (**b,d,f,h,j**) Gram staining of the bacterial morphology at 0.75, 0.375, 0.188, 0.094, and 0.047 ppm NaOCL concentrations, respectively. (**k**) Fluorescent staining of the bacterial morphology using water. (**l**) Gram staining of the bacterial morphology using water.

**Table 2.** Distribution of the decrease in the number of bacteria by treatment group.

| Treatment Groups | n | Minimum | Maximum | Mean | Std. Deviation | *p* |
|---|---|---|---|---|---|---|
| Soil 30% + Chlorine 0.75 | 10 | 41 | 56 | 49.4 | 5.80 | 0.147 |
| Soil 30% + Chlorine 0.375 | 10 | 51 | 71 | 61.7 | 6.48 | 0.929 |
| Chlorine 0.75 + Soil 30% | 10 | 53 | 124 | 86.7 | 22.06 | 0.227 |
| Soil 30% + Chlorine 0.1875 | 10 | 78 | 103 | 93.3 | 8.62 | 0.495 |
| Chlorine 0.375 + Soil 30% | 10 | 36 | 300 | 126.8 | 67.46 | 0.660 |
| Chlorine 0.1875 + Soil 30% | 10 | 104 | 208 | 153.6 | 37.38 | 0.125 |
| Soil 30% + Chlorine 0.09375 | 10 | 188 | 298 | 245.5 | 37.40 | 0.965 |
| Chlorine 0.09375 + Soil 30% | 10 | 170 | 348 | 271.2 | 60.00 | 0.003 |
| Soil 30% + Chlorine 0.046 | 10 | 276 | 350 | 307.4 | 23.95 | 0.318 |
| Chlorine 0.046 + Soil 30% | 10 | 280 | 420 | 348.5 | 54.16 | 0.370 |
| Soil 30% + Water | 10 | 300 | 500 | 419.0 | 78.80 | 0.092 |
| Water + Soil 30% | 10 | 300 | 500 | 419.0 | 78.80 | 0.125 |

Bacteria before and after treatment are shown in Figure 2.

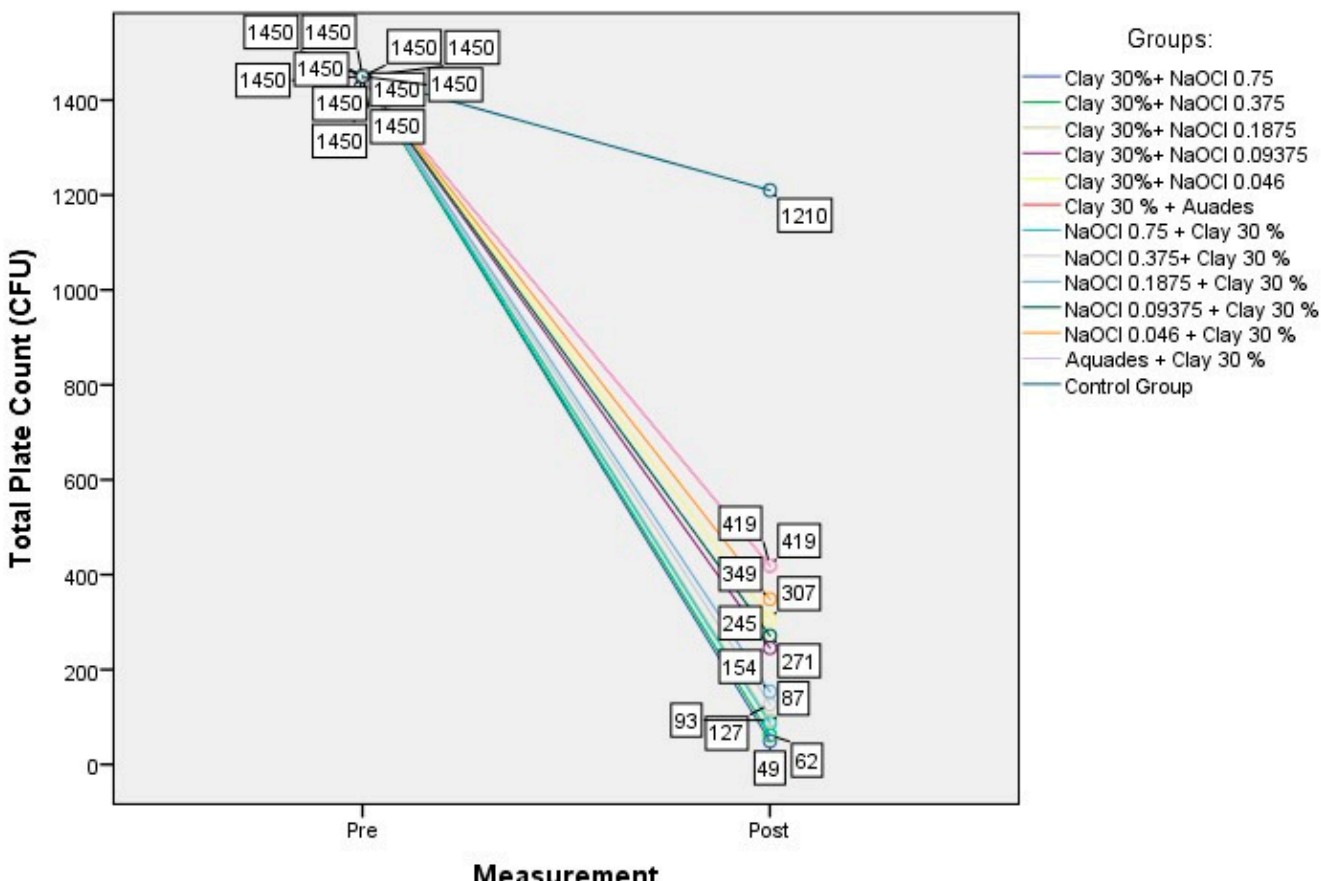

**Figure 2.** Bacteria reduction before and after treatment.

Figure 2 demonstrates that the number of colony-forming units (CFUs) of bacteria was 1450 before receiving treatment. Following the administration of therapy, each combination could produce a reduction in the number of bacteria, the extent of which ranged from 49 CFUs to 419 CFUs. The number of CFUs of bacteria only dropped to 1210 in the group that served as the control. The combination of these treatments achieved a sizeable drop in the bacterial population. Table 3 demonstrates how effectively the treatment reduced the total number of bacteria in the environment. The data presented in the table demonstrate that there was a reduction in the number of bacteria that were able to survive as a result of the interaction between the soil concentration and the NaOCl dose, as well as the sequential application of these two variables. The treatment that consisted of a mixture of 30% dirt and 0.75 ppm chlorine was the one that was able to lower the number of bacteria by the greatest amount out among all the available options. When syringe waste from hospitals was treated with soil containing 30% chlorine and 0.75 ppm chlorine, the amount of bacteria that was removed was 1161 CFU lower than that of the control group. The use of the combined treatment caused a 96% reduction in the bacterial count. The ability of the combination of the soil concentration and NaOCl dose to clean bacteria was ordered from large to small, as shown in the table, and the combination treatment consisting of water and soil with a concentration of 30% was the treatment that exhibited the lowest variety in terms of cleaning bacteria. This treatment was able to reduce the number of bacteria by 791 CFU. The combined treatment had the potential to kill 93% of the germs.

According to the findings of [32], reducing the concentration of NaOCl to 0.2 mg-Cl$_2$/L and using a contact period of 30 min enabled the researchers to cut the number of bacteria by 90 per cent. Another research group [19] found that the suggested level of NaOCl, which was 0.5 mg/L, was less effective in reducing the number of *Escherichia coli* isolated from two wastewater treatment plants; they were still present in the water. A treatment

in which the concentration of NaOCl was 1.5 mg/L was found to be more successful in eliminating *Escherichia coli* bacteria. After 180 min of exposure, another investigation with a concentration of 1.5 mg/L demonstrated a deadly effect (100%) on *Escherichia coli*, *Staphylococcus aureus*, and *Klebsiella pneumoniae* [32].

**Table 3.** Effect of various combination treatments on bacteria.

| Treatments | B | Std. Error | t-Value | *p* | Partial Eta Squared |
|---|---|---|---|---|---|
| Soil 30% + Chlorine 0.75 | −1161 | 20.57 | −56.41 | <0.001 | 0.9645 |
| Soil 30% + Chlorine 0.375 | −1148 | 20.57 | −55.82 | <0.001 | 0.9638 |
| Chlorine 0.75 + Soil 30% | −1123 | 20.57 | −54.60 | <0.001 | 0.9622 |
| Soil 30% + Chlorine 0.1875 | −1117 | 20.57 | −54.28 | <0.001 | 0.9618 |
| Chlorine 0.375 + Soil 30% | −1083 | 20.57 | −52.65 | <0.001 | 0.9595 |
| Chlorine 0.1875 + Soil 30% | −1056 | 20.57 | −51.35 | <0.001 | 0.9575 |
| Soil 30% + Chlorine 0.09375 | −965 | 20.57 | −46.88 | <0.001 | 0.9495 |
| Clorin 0.09375 + Chlorine 30% | −939 | 20.57 | −45.63 | <0.001 | 0.9468 |
| Soil 30% + Chlorine 0.046 | −903 | 20.57 | −43.87 | <0.001 | 0.9427 |
| Chlorine 0.046 + Soil 30% | −862 | 20.57 | −41.88 | <0.001 | 0.9375 |
| Soil 30% + Water | −791 | 20.57 | −38.45 | <0.001 | 0.9267 |
| Water + Soil 30% | −791 | 20.57 | −38.45 | <0.001 | 0.9267 |

Using a single dosage of NaOCl with a concentration of 0.75 ppm was less efficient in lowering the number of bacteria than using the overall treatment. A combination of soil with an absorption mechanism and the capability to kill bacteria is required, owing to the ability of the activated metal content to boost the treatment's ability to decrease the total number of bacterial colonies (Fe, Zn, Cu, Pb). With a reduction efficacy of 93%, the combination of 30% soil and 0.75 ppm NaOCl was the treatment that could most successfully cut down on the number of bacteria.

The presence of soil alone can lessen the number of bacteria. However, the potential of this approach still needs to be fully realized. Researchers have offered various alternatives for enhancing the capability of the soil, one of which consists of administering a treatment that combines NaOCl and other chemicals to make the soil more effective. The number of bacteria in an area can be decreased via the physical and chemical effects of the soil. The capacity of soil particles to lower the number of bacteria is affected physically by the nature of the soil particles.

In contrast, the ability of metal components in the soil to kill bacteria is affected chemically by the presence of metal components in the soil. The addition of NaOCl results in an enhancement of this capability. The process that causes the reduction in the number of bacteria is related to NaOCl. Specifically, active chlorine compounds are what cause harm to the cells of the bacteria. Mixed NaOCl soil has the advantage of enabling users to lower the amount of NaOCl necessary to achieve the desired result, that is, reducing the population of *B. cereus* bacteria.

An unambiguous system for managing clinical waste is required in Indonesia, as in the majority of other nations with emerging economies; moreover, there needs to be more data available on managing medical waste in that country [33]. It should not come as a surprise that medical waste management has received less research attention and has been treated with less urgency given the current state of affairs in the country, in terms of both health concerns and the limited availability of resources. The development of dependable records of the quantity and characteristics of healthcare wastes, as well as the management methods to adequately dispose of these wastes, have remained challenging, and it is believed that several hundreds of tons of healthcare waste have been stored directly in waste dumps and neighboring environments, typically together with non-hazardous waste. This is

a problem because healthcare waste can risk human health and the environment. The careless disposal of waste from medical services poses risks to both individual health and the health of the environment by polluting air, land, and water resources. In the absence of adequate supervision, waste generated by medical services can pose a threat that is significantly more dangerous than the sickness itself. The community's overall health should be protected by hospitals and other medical care facilities. According to the findings of an inquiry into the management of clinical waste in three emergency clinics in Semarang, Indonesia, the clinical waste administration choices made by the medical clinics did not adhere to the established standards. A robust study of waste collection procedures revealed that open dump locations were frequently selected; however, burning did not occur in any of the examined clinics or institutions. However, none of the three medical clinics kept records of when waste was removed or how old it was, nor did they separate waste into stamped or variety-coded boxes according to the various waste streams. Furthermore, it was discovered that open dump locations were not even constructed or treated, which exposed them to the inherent risks of contamination.

Hospitals, clinics, health care facilities, and COVID-19 centers are the primary generators of COVID-19 medical waste because these establishments produce a variety of COVID-19 medical waste in the form of diagnostics waste, research and laboratory waste, infectious waste, cytotoxic waste, chemical and radioactive wastes, drugs, some other medicinal waste, etc. When not disposed of appropriately, most of this trash contains hazardous elements that represent a risk to the general population and the environment. Because of this, the initial phase of its control begins with the COVID-19 patient care centers and hospitals from which it originates. For this reason, this garbage should be sorted adequately, and it must be collected in separate bags or bins that have been allocated to accomplish this particular aim. Before being transported to facility centers, the bags containing syringe waste should first be treated with a combination of 30% soil and 0.75 ppm NaOCL and then placed inside double-layered plastic bags. Currently, the COVID-19 trash bags that have been separated are being maintained in storage areas, from which they are collected according to a prioritized schedule or within predetermined time limitations. Before its transfer to standard COVID-19 waste treatment and disposal facility centers, this temporary storage location and the vehicles that are utilized for its transportation are required to undergo routine disinfection to reduce the risk of infection among workers. An appropriate method of appropriate disinfection must be chosen, taking into account the quantity and kind of the waste to be treated, as well as its cost and level of upkeep. Processes that take place at temperatures over 1000 degrees Fahrenheit, such as incineration and pyrolysis [34] are two methods that can be utilized to provide treatments on a massive scale for this trash and eliminate infective surfaces. After that, the waste undergoes microwave treatment, using steam disinfection technology that operates at a high temperature.

The previous results and findings presented in the *Systematic Literature Review and Bibliometric Study of Waste Management in Indonesia in the COVID-19 Pandemic Era* [9] highlighted some points to be considered as future recommendations. From the standpoint of sustainability, the authors urged that once the COVID-19 pandemic is over, we must investigate additional waste handling processes in order to achieve a sustainable environment. Among these procedures are preventive design. This handling process focuses on preventing and/or minimizing waste generation by maximizing resource potential. Another procedure is alternative waste management—this type of waste management goes beyond recycling to include upcycling (recycling with the addition of value). Reviews of the research on waste management, as well as original studies on the pyrolysis of solid wastes (medical, plastic, etc.), are important from a scientific standpoint, particularly for Indonesian academics, in closing the research and publication gaps in the area of waste management in Indonesia during the COVID-19 pandemic era. This study has responded to this need.

## 4. Conclusions

The vast quantity of medical waste produced by hospitals, including worn facemasks, gloves, gowns, syringes, and other items, has been primarily categorized as hazardous since these waste products carry infectious pathogens such as viruses. The possibility of viral contamination may grow as a result of improper management of the aforementioned wastes along with typical municipal garbage, which may also enhance the likelihood of virus transmission. MW constitutes a significant portion of infectious wastes, and is potentially dangerous because it may be resistant to treatment and possess high pathogenicity or a heightened ability to cause disease. Therefore, facilities providing medical care have a responsibility to be aware of the potential dangers involved in handling infectious waste and they must adhere to the strictest possible guidelines regarding its disposal and transportation. In today's healthcare system, educating staff, patients, and community members about the proper management of infectious waste is necessary. As a result, its efficient management is of the utmost importance. This can only be accomplished by implementing several distinct procedures, such as appropriate identification, collection, segregation, storage, transportation, treatment, and final disposal.

In general, just like the majority of other developing countries, Indonesia has lacked the technology and institutional capacity necessary to effectively manage medical waste to enhance the protection of human life and the environment from the health hazards that can arise from the improper management of hazardous waste. Therefore, the use of a combination treatment consisting of soil (at a percentage of 30 per cent) and chlorine (at a concentration of 0.75 parts per million) was shown to be the most effective treatment in terms of reducing the total number of microbial colonies (at a percentage of 93 per cent), which may be the first step that can be taken in practice. Because this method was only tested on sharp medical waste (syringes), we recommend conducting additional research, thus increasing the potential use of soil as a single treatment for a variety of bacterial species.

**Author Contributions:** Conceptualization, M.M. and I.R.; methodology, M.M.; software, M.M.; validation, M.M. and I.R.; formal analysis, M.M.; investigation, M.M.; resources, M.M.; data curation, M.M.; writing—original draft preparation I.R.; writing—review and editing, I.R.; visualization, M.M.; supervision, M.M.; project administration, I.R.; funding acquisition, M.M. All authors have read and agreed to the published version of the manuscript.

**Funding:** This research was funded by the Ministerial of Health Polytechnic Semarang, Indonesia and The APC was funded by Ministerial of Health Polytechnic Semarang, Indonesia.

**Institutional Review Board Statement:** Not applicable.

**Informed Consent Statement:** Not applicable.

**Data Availability Statement:** Not applicable.

**Acknowledgments:** The authors would like to express their gratitude to the Ministerial of Health Polytechnic Semarang, Indonesia, for providing funding for the study, as well as the Rector of Lambung Mangkurat University Banjarmasin, Indonesia, for assisting in facilitating the researchers' collaboration and providing funding for the study.

**Conflicts of Interest:** The authors declare no conflict of interest.

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
