# Peer review of "Novel Dose of Natrium Chloride and Soil Concentration in Reducing Medical Waste Bacteria before Incineration"

_applsci, doi:10.3390/app13042119_

Round 1

Reviewer 1 Report

Specific comments:
1. Please make sure that "Natrium Chloride" in the text is correct? "Natrium Chloride" or "Sodium hypochlorite" for example, Line2, 62, 64, etc. Please check the full text.
2. In the first paragraph of the introduction, the authors highlighted that the “ the adverse effects related to the increase of solid waste and decreased recycling efforts have also become apprehension” caused by COVID 19, but the bacterial strain “Bacillus cereus” selected in the subsequent trial did not show an association with COVID 19.
3. In the introduction, "medical waste (MW)" (line30) appeared first, followed by "medical waste" and then "MW". Please check the full text and unify the expression. Similarly, line62 and line64, repeated abbreviations. Please check the full text.
4. Line66 68: “The ability of soil to reduce B. cereus bacteria in syringe medical waste has been carried out and proven [13], but its effectiveness in combination with various doses of NaOCl disinfectant has not been examined.”What is the relationship between the "syringe medical waste" mentioned here and the aforementioned medical waste caused by COVID 19? You are not arguing that the emergence of COVID 19 has caused an increase in syringe care waste or caused syringe care waste to cause greater harm. So why are you looking at syringe medical waste? Why not a mask or protective clothing or something? Please clarify.
5. Why choose the combination of soil and disinfectant to reduce bacteria? "Since dumping into the soil and using Sodium chloride (NaOCl) are widely practised in developing economic conditions countries " in the introduce of line62 63 , the evidence is insufficient.
6. Line71 72: Why choose "Bacillus cereus bacteria" as the test object? Is it related to COVID 19? Please clarify. In addition, the author only selected one kind of bacteria, Bacillus cereus, as the object, can the conclusion of "reducing the total number of microbial colonies (93%)" be drawn?
7. Line81 83: "the chemical content of the soil was able to reduce the number of bacteria, including Fe (levels of 11.772 ppm), Zn (levels of 0.105 ppm) and Cu (levels
of 0.018 ppm)." It is more appropriate to appear in the introduction.
8. Line86: "Syringes that have been contaminated with Bacillus cereus bacteria for 8 hours" why only 8 hours? The actual MW should last more than 8 hours from being discarded to being disposed of. As we know, bacteria are microorganisms with strong reproductive ability, which can reproduce rapidly in a short time. If you only choose MW contaminated for 8 hours, can you truly reflect the efficacy of the fungicide? If not, then the data from the trial are not reasonable. If there is evidence, please explain it briefly in the text.
9. Line87--88: "NaOCl (30 minutes)", "30% andosol soil (2 minutes)". Why is the processing time 30minutes and 2minutes, based on what?
10. Are there duplicate samples? How many replicate samples per treatment? What's the total sample? Please explain.
11. Line109--112: “Soil is potential for reducing the number of bacteria. This is because Zn content in soil can reduce the number of Escherichia coli bacteria by 98% with a contact time of 2 hours and 100% of Pseudomonas aeruginosa colonies within 4 hours[7].”This is a description of the literature, not an explanation of why soil reduces bacteria.
12. Line112--114: Why Ferric ion iron (Fe3+) can reduce the number of Escherichia coli, please explain.
13. Line114: "Soil can degrade (bacteriostatic) up to kill bacteria (bactericide)." The argument is not sufficient to draw this conclusion.
14. Line104--125: These two paragraphs would be better placed in the introduction for a review of the antibacterial effects of soil and NaOCl.
15. Line115--117: Is it appropriate to explain terminology here? What does this have to do with the whole discussion of this paragraph?
16. The results and discussion section is more about the description of the results and lacks discussion.
17. Line210--228: Not appropriate for the results and discussion section, more like research background.
In general, the article is not rigorous in logic and is poorly expressed in many
places. It is more like a draft than a qualified journal article.

Author Response

Dear Reviewer, 

We have tried to do our best to improve the paper as suggested.

We would like to thank for your valuable comment  on making the paper is publishable. 

Reviewer 2 Report

In this study, variations in the dose of disinfectant combined with andosol soil to reduce the number of bacteria found in medical waste produced from hospitals in Semarang City, Indonesia was analyzed. The test results found that the combination variation between soil (30%) and chlorine (0.75 ppm) was the most effective in reducing the total number of microbial colonies (93%). The manuscript presents interesting results, which are relatively well organized and systematized, but the novelty and practical applicability of this study should be highlighted more. Also, it will be useful to include some information regarding the economic impact of the work. In my opinion, this manuscript should be published after minor revision.

Here is a list of my general comments:

·         The novelty, economic impact and practical applicability of this study should be highlighted more.

·         Define abbreviations at first mention. Abbreviations should be defined at first mention and then through the text use only the abbreviations not the full name and use the same abbreviation through the manuscript.

·         Specific comments:

o    Line 62: The NaOCl is Sodium hypochlorite not sodium chloride. Sodium chloride is NaCl. Please correct.

o    Line 64: Use only abbreviation for NaOCl, because it was already introduced in line 62.

o    Line 100: Write first letter in Zink with lowercase.

o    Line 118: Use only abbreviation for NaOCl.

o    Line 198: Write 2 in Cl2 in subscript.

o    Line 232: The abbreviation Medical wastes (MW) is already introduced. Please, use only abbreviation.

Author Response

(The authors gave the same response as above.)

Round 2

Reviewer 2 Report

In the revised version the authors accepted all comments and made the necessary changes, so I proposed to accept this manuscript for publication.